# The Effects of Latent Attitudinal Variables and Sociodemographic Differences on Travel Behavior in Two Small, Underdeveloped Cities in China

**Gang Cheng [1,2,\*], Shuzhi Zhao [1] and Jin Li [1]**

[1]  College of Transportation, Jilin University, Changchun 130000, China; zhaosz@jlu.edu.cn (S.Z.); li_jin@jlu.edu.cn (J.L.)
[2]  College of Engineering, Tibet University, Lhasa 850000, China
[\*]  Correspondence: fengyunleng@163.com

**Abstract:** In small, underdeveloped Chinese cities, residents have few options for transportation and travel service problems have not received enough attention from the authorities. This study examines residents' preferred mode of travel in Lhasa and Yushu, China, two small and underdeveloped cities in which travelers tend to be economically disadvantaged. Travel data from different regions was analyzed to explore their commonalities. A structural equation model with latent variables is proposed to capture the heterogeneity not observed in the selection process. Results indicate that four of the six latent variables—preference for comfort, preference for reliability, preference for convenience and safety consciousness—are more helpful than preferences for flexibility and environmental awareness in explaining transportation utility, which could reflect residents' travel behavior. Based on the results, respondents were divided into five groups with similar travel preferences through the k-means clustering method. The findings show that ensuring high comfort and convenience and moderate safety and reliability is conducive to increasing residents' use of public transport. Furthermore, an examination of residents' sociodemographic differences reflects that, in the future, active transport demand management should focus on trying to satisfy the preferences of female, low-income and elderly travelers.

**Keywords:** economically disadvantaged residents; travel behavior; latent attitudinal; structural equation model; difference analysis

## 1. Introduction

In China, uneven production and development have taken place across different regions because of disparate access to science and technology. Particularly since the reform and opening up, the gap between underdeveloped and developed regions has become increasingly obvious. Underdeveloped areas refer to those regions with a certain economic strength but a potential gap (e.g., disposable income and available transportation resources), when compared to developed regions such as Nanjing, Shanghai, Guangzhou and Beijing. Referring to the international poverty line standard, proposed by the Organization for Economic Cooperation and Development, combined with the actual situation in China, the poverty rate in big cities is defined as the income level of 50% of the regional median disposable income per capita. In 2016, the lower-middle-income line in Nanjing was a monthly per capita income of 3074 Chinese Yuan [1]. In most small, underdeveloped cities in China, with a population of less than 500,000 [2], most residents belong to the economically disadvantaged group, including lower-middle-income and low-income groups [3], which both enjoy the Chinese preferential policies.

Over time, urbanization has changed typical travel modes: from walking and cycling to various motorized means of transport including by car, motorcycle, electric bicycles (e-bikes) and public transportation, even in underdeveloped areas [4,5]. The cost of private cars is too high for economically disadvantaged residents to afford and, as such, they depend heavily on non-motorized travel modes and public transport. Most of their trips are driven by subsistence activities and are less likely to be for maintenance and/or discretionary activities. For low-income residents from families who earn the national minimum living allowance, managing the family budget to balance the cost of travel and all other expenses, may add expenditure pressure and reduce their ability to participate in additional activities or travel beyond a walking distance from their place of residence [6]. The suburbanization of low-wage jobs and the increasing suburbanization of economically disadvantaged residents mean that public transport cannot provide sufficient service to reach multiple employment opportunities disbursed throughout the region [7,8]. Modes of transportation are also limited in small and underdeveloped cities; subways and light rail transit, for instance, are often excluded because of inadequate transportation infrastructure. Most residents depend heavily on non-motorized transportation such as walking, cycling and e-bikes as their primary mode of travel. The public transportation modal split in underdeveloped areas is lower than in developed cities due to the lack of a complete public transport service system [9,10]. Through the "poverty alleviation policy," the Chinese government has begun to encourage the in-situ urbanization and economic development of underdeveloped areas [11]. This national policy has quickly enhanced urban motorization and has stimulated, especially, a rapid growth in the use of private cars, causing a series of traffic problems in small, underdeveloped cities. The contextual factors of small, underdeveloped cities—such as economic development, urban form and infrastructure construction—are also different from those of large cities [12,13]. Therefore, the behavior of residents in these cities is unique.

Existing knowledge of travel behaviors and their underlying causes in small, underdeveloped cities is inadequate for guiding the development of low carbon transport. Research on travel behavior in this kind of area has not received enough attention and related studies are scarce. In addition, few previous studies explore the relationship between travel behavior and latent attitudinal effects in economically disadvantaged groups. This study aims to address this knowledge gap by examining residents' choice of travel mode in two small, underdeveloped cities. Both large and small cities face growing transportation problems and research is necessary to guide the development of low-carbon transportation policies and technology to avoid more serious traffic congestion and pollution in the future.

This study is primarily intended as an analysis of how residents' latent attitudes affect their travel behavior and also examines their sociodemographic differences in small, underdeveloped Chinese cities. It provides an interesting case study in a Chinese context to highlight the ordinary nature of the two small cities—Lhasa and Yushu—which are both underdeveloped and have limited transportation resources and economically disadvantaged oriented travelers.

## 2. Literature Review

Studies conducted in many cities have found that abstract psychological constructs, such as attitudes, values, perceptions, affects and desires, are integral to an individual's choice of travel mode [14–17]. While the role of instrumental factors in determining choice of travel mode has been recognized most prominently, psychological and social factors have also received considerable attention. In some cases, attitude may be more advantageous than objective measures, such as time and costs, in predicting mode choice [18].

In their recent report, Paulssen et al. [19] attempted to examine the influence of values on travel mode choice behavior. They found that different personal values will have differences in alternative traffic mode attributes, which in turn impact modal choices. They constructed an integrated choice and latent variable model of travel mode choice and allowed for hierarchical relationships between latent variables and flexible substitution of patterns across modal alternatives.

Heinen et al. [20] showed that, compared to observable level-of-service attributes, latent attitudinal variables—awareness, direct trip-based benefits and safety—have greater influences on mode choices. Findings show that attitudinal factors and other psychological factors have a relatively stronger impact than observable variables (e.g., convenience, low cost, health benefits) on an individual's choice to commute. Grdzelishvili and Sathre [21] studied the relationship between urban transport attitudes and behavior in Tbilisi. The authors demonstrated that the issues of time, comfort and safety were the most important factors in survey respondents' preference to use a private car and avoid using public transport, further evidencing that attitudes play an important role in the choice of travel mode. Various attitude-behavior models have been developed that attempt to identify and assess the various moderating factors that affect attitude-behavior consistency [22,23].

The relationship between latent variables and travel behavior can be analyzed through structural equation modeling (SEM), which is an extremely flexible linear in parameters multivariate statistical modeling technique [24]. Golob [25] proposed a joint model of attitude and behavior to explain how both mode choice and attitudes regarding the San Diego I-15 Congestion Pricing Project differed across the population. Results show that some personal and situational explanations of opinions and perceptions are attributable to mode choices. SEM was used in the study to identify the attitudes of travel behaviors and the causal relationships between traveler's socioeconomic profile and traveler attitudes, simultaneously. Outwater et al. [26] extracted six attitudinal factors, three of which were used to divide the ferry market into eight segments in San Francisco. These market segments were used to estimate 14 alternative models of stated preference model selection, which recognized that mode choices are different for market segments that are sensitive to travel stress or those with a desire to help the environment. Much of the research in the last two decades suggests a stronger causal link between choices and attitudes. Previous studies emphasized that analyzing attitude and other data was useful for understanding respondents' intentions but caution was recommended regarding survey wording and measuring behavior intentions [27]. Popuri et al. [28] considered the impact of traveler attitudes and perceptions through a survey of public transport choice in the Chicago area. A factor analysis methodology was used to condense scores on 23 statements related to daily travel into six factors and the factor scores on these six dimensions were used in conjunction with traveler socioeconomics, travel times and costs, to estimate a binary logistic regression of public transit choice. Overall, the results of this study suggest that attitudinal factors improved the intuitiveness and goodness of fit of the model. This brief literature review shows that travel behavior can be suitably explained by these latent variables. In this research, the primary intention was to study the effects of latent attitudes and sociodemographic differences on travel behavior and to understand what actions are needed to fulfill respondents' psychological preferences.

Researchers have employed various case studies and methods to evaluate the impacts of individual attitudes toward less tangible attributes such as comfort and convenience on mode choices. By means of 1500 standardized telephone interviews, Haustein [29] assessed the mobility behavior of the elderly and its possible determinants, including infrastructural, sociodemographic and attitudinal variables. The study provides a more comprehensive understanding of the diverse lifestyles, attitudes, travel behaviors and needs of the elderly and helps reduce car use. Cheng et al. [30] found that including the activity participation endogenously in SEM is advantageous to understanding travel behavior, which has complex relationships among sociodemographic, accessibility, activity participation, trip generation and mode choice. Findings in this study reveal the effects of accessibility variables on activity participation and travel behavior in which population density measure has more ubiquitous effects. This brief review of the literature suggests that travel behavior can be better explained by both potential and observational variables. As is evident, however, the mobility of economically disadvantaged commuters in underdeveloped cities has not received enough attention.

In addition, as mentioned by Cheng et al. [31] most of these studies are focused on relatively homogeneous groups of people such as university students, the elderly, commuters and immigrants in big cities. Furthermore, there are few contributions on economically disadvantaged commuters in small, underdeveloped cities. Despite the increasing attention devoted to the study of the model choice of economically disadvantaged residents, the current literature has certain gaps that have motivated this research.

The contributions of this paper are threefold. With few exceptions, previous studies have used paper-based intercept surveys. One exception worth noting is a study conducted by Barajas et al. [32] who used mixed-methods including surveys to compare the travel patterns of economically disadvantaged immigrants living in the San Francisco Bay Area with that of other groups. In our study, the sample presented higher variability because the survey of respondents included two small underdeveloped Chinese cities, reducing potential bias in the results.

In addition, much of the previous work on economically disadvantaged residents' travel has relied on national surveys and qualitative analyses, which have focused mainly on large cities in the developed countries of Europe as well as the U.S., while research focused on Chinese urban residents is relatively scarce [33,34]. Few have considered small, underdeveloped, Chinese cities, in which the behavior and the underlying motivations might be different. In this paper, we conducted surveys in two small, underdeveloped Chinese cities, which are well representative of disadvantaged population groups and are able to speak to broader patterns in the population.

Finally, most previous studies in this field have used traffic surveys, telephone surveys and qualitative interviews to understand particular influences on travel behaviors [35,36]. Most research has included overall income as a variable examining sociodemographic differences instead of personal disposable income, which has higher accuracy [37]. In this research, we extend these methods, including interviews and original surveys. The analysis in this research explicitly considered latent attitudinal effects on travel behavior.

## 3. Data Sources and Descriptive Statistics

Data was collected from a detailed survey which includes items concerning existing travel behavior, prevailing attitudes, beliefs, the experiences of inhabitants and behaviors pertaining to travel in Lhasa and Yushu. Each city is representative of areas in which little published research exists on attitudes towards different transport modes. Lhasa is the capital of the Tibet Autonomous Region in the Southwest and Yushu is in Jilin province in the Northeast part of China. A common feature of the two cities is that they are both underdeveloped and available transportation resources are very limited. The following section presents a brief summary of the main characteristics of the two cities, especially in terms of population, income distribution and some mobility patterns.

Lhasa is one of the most impoverished areas with pilgrim-oriented travelers and the city's population is around 300,000. The majority of residents are economically disadvantaged who require government policies to alleviate poverty. Due to a serious lag in the development of public transport, the number of private cars has increased rapidly in recent years, exacerbating the impact of terrain and traffic pressure. Like Yushu, Lhasa has a routine bus system with a simplex service function. There are 33 bus routes in service and around 40 km of pedestrian passageways in the central district of the city. There is a free ticket policy for all registered people aged 60 and above. The Yushu metropolitan area, which is home to around 260,000 of people, is smaller than that of Lhasa. There are approximately 0.83 households per vehicle, so car ownership is also notably lower than in Lhasa. The main means of transportation are e-bikes, private cars and motorcycles. In fact, the percentage of private car per household in Lhasa is one of the highest in underdeveloped areas due to the serious lag in public transport development. In Yushu, car concentration is also relatively high compared to other small cities. The annual growth rate of motor vehicles (e.g., motorcycles and private cars) is much higher than the growth rate of public transportation in both cities. Specifically, in 2017, the number of motor vehicles in Lhasa exceeded 226,000. Public transportation has not been able to meet rising travel demands

in small, underdeveloped cities such as these, due to a lack of efficient management and operation. Only 19.76% and 16.53% of journeys in the metropolitan areas of Lhasa and Yushu, respectively, were made via public transportation. Pedestrian traffic still accounts for a large modal split because of low disposable income; over 30% of journeys were on foot in two cities. Moped transportation, such as motorcycles and e-bikes, is very appealing because it is inexpensive and can provide door-to-door travel. However, there are not enough separate lanes for moped transportation and this worsens congestion and increases accidents.

By including two somewhat different, small Chinese cities, we verified whether our primary findings held across both cities in the sample. We were also able to ascertain the extent to which previous research findings on travel behavior and psychological preferences in Lhasa and Yushu are consistent with other small, underdeveloped cites, thus providing valuable insight into the generalizability of our findings. This study also tries to determine how to guide governments of small cities to achieve continuous low-carbon travel behavior in underdeveloped areas.

To explore the travel behavior in small, underdeveloped Chinese cities, we conducted intermittent observation of residents in Lhasa and Yushu in 2010 and 2014, respectively. In 2015 and 2016, questionnaires were used to gather information from respondents for statistical analysis. A paper-based questionnaire was used mainly for elderly residents in combination with interviews. Online questionnaires were sent through an Internet questionnaire survey platform, by email and chat software (e.g., WeChat and QQ) using various methods. In Yushu, the local government assisted us to approach school teachers who distributed the surveys to the parents of students, some of whom returned the completed questionnaires. This is a common process for conducting surveys in small cities in China. According to the typical travel and localization characteristics of Chinese urban residents, we discovered possible problems and tested the validity of the scale through preparatory research in small samples. The formal scale included three parts: latent attitudinal effects of travel, statistics of household heterogeneity and a one-day travel record including the detailed information about respondents' mobility information on the day of the survey. Taking a household as a unit, a total of 1000 questionnaires were assigned in the residents' travel survey, of which 856 were valid, giving an effective return rate of 88.1% and a recovery rate of 77.06%. Of the total 856 households who completed the questionnaire: 403 were completed in Lhasa, with 1358 (47%) respondents and 453 in Yushu, with 1732 (53%) respondents. Demographic differences between cities are the result of sampling and are representative of the demographics of the city, overall. In order to balance the population differences between cities, our analysis does not focus on the differences between cities but rather on their similarities.

Apart from the observable variables (e.g., gender, mode of transportation, family members), the attitudinal variables (e.g., safety consciousness, comfort preference, convenience preference) were also collected. These attitudinal variables were measured on a five-point Likert scale from "not important at all" to "very important." Environmental awareness and safety awareness are also associated with individual behavior and were also measured on a five-point Likert scale from "never" to "always." Items for the latent variables are shown in Table 1.

The sociodemographic items of the survey are shown in Table 2 and include individual characteristics (e.g., gender, age), household characteristics (e.g., family size, motorcycle ownership, car ownership) and trip characteristics (e.g., travel mode share, travel purposes).

During the preparatory research, we established that lanes designed for bicycles are scarce in the two cities. Accordingly, we combined bicycle transport into pedestrian transport and available options were grouped into these five categories: (i) pedestrian transport for short-distance trips, including walking and bicycles; (ii) moped transportation, providing door-to-door travel, including motorcycles and e-bikes; (iii) public transportation, including buses, commuter buses, school buses and county transportation vehicles; (iv) taxi transportation; (v) private transportation including private vehicles and network reservation vehicles. These transportation modes were used to evaluate the traffic utility of the respondents.

In addition, the careers of the respondents were divided into two categories, including stable career and non-stable career. The stable career option includes student, soldier, government worker and institution worker, while the non-stable career includes restaurant waiter, farmer, hourly worker, babysitter and so forth. This is mainly because of the remnants of the traditional socialist danwei system in the two cities: groups with stable careers tend to benefit much more from the local government.

**Table 1.** Latent Variable Items.

| Latent Variable | Item | No. |
|---|---|---|
| Comfort preference (A) | I prefer to be able to move around while traveling | A1 |
| | I prefer to be able to rest on a seat while traveling | A2 |
| | I prefer to be able to open a window while traveling | A3 |
| | I prefer traveling in a quiet environment | A4 |
| Convenience preference (B) | I prefer not having to transfer while traveling | B1 |
| | I prefer not having to wait for transit while traveling | B2 |
| | I prefer to be able to arrive at a destination quickly | B3 |
| | I prefer to be able to carry baggage while traveling | B4 |
| | I prefer having various options for payment available while traveling | B5 |
| Reliability preference (C) | I prefer having little variation in daily travel time | C1 |
| | I prefer to be able to know the arrival time of the transit mode | C2 |
| | I prefer to be able to judge or control the travel time | C3 |
| | I prefer to know that the mode of transportation will wait for me | C4 |
| Flexibility preference (D) | I prefer to be able to shop while traveling | D1 |
| | I prefer to be able to complete personal-related affairs while traveling | D2 |
| | I prefer to be able to complete family-related affairs while traveling | D3 |
| | I prefer having multiple routes or roads available while traveling | D4 |
| Safety consciousness (G) | I prefer to be able to adhere to the speed limit when driving | G1 |
| | I prefer to be able to wear a helmet or seatbelt | G2 |
| | I prefer to be able to avoid running through a red light | G3 |
| | I prefer to be able to avoid infringing the right of way | G4 |
| Environmental awareness (H) | I believe that reducing the use of private cars is conducive to improving air quality | H1 |
| | I would change my travel mode if it would be helpful for the environment | H2 |
| | I believe public transit is beneficial to protecting the environment | H3 |
| | I do not think there are environmental pollution problems in the city | H4 |

**Table 2.** Sociodemographics of Analytic Samples.

| Variable | Coding | Categories | Respondents | Sample % |
|---|---|---|---|---|
| Household size | Size | Three or fewer people (Size 1) | 627 | 70.7 |
| | | More than three people (Size 2) | 260 | 29.3 |
| Car ownership per household | Car | One car or more (Car 1) | 378 | 42.6 |
| | | No cars (Car 2) | 509 | 57.4 |
| Motorcycle ownership per household | Moped | One motorcycle or more (Moped 1) | 358 | 40.4 |
| | | No mopeds (Moped 2) | 529 | 59.6 |
| Gender | Gender | Female (Gender 1) | 1551 | 51.3 |
| | | Male (Gender 2) | 1473 | 48.7 |
| Age | Age | <18 years old (Age 1) | 599 | 19.8 |
| | | 19–34 years old (Age 2) | 502 | 16.6 |
| | | 35–54 years old (Age 3) | 1026 | 33.9 |
| | | 55–64 years old (Age 4) | 522 | 17.3 |
| | | ≥65 years old (Age 5) | 375 | 12.4 |
| Career | Career | Stable career (Career 1) | 1417 | 46.9 |
| | | Non-stable career (Career 2) | 1607 | 53.1 |
| Number of trip chains | Chain | Simple chain (Chain 1) | 1772 | 58.6 |
| | | Complex chains (Chain 1) | 1252 | 41.4 |
| Travel purposes | Purpose | Work (Pur 1) | 1279 | 42.3 |
| | | School (Pur 2) | 459 | 15.2 |
| | | Business (Pur 3) | 338 | 11.2 |
| | | Life-related (Pur 4) | 611 | 20.2 |
| | | Entertainment (Pur 5) | 186 | 6.2 |
| | | Shopping (Pur 6) | 151 | 5.0 |
| Personal disposable income | Income | RMB ≤ 1499 (Income 1) | 1305 | 43.2 |
| | | RMB 1500–2499 (Income 2) | 943 | 31.2 |
| | | RMB 2500–3500 (Income 3) | 566 | 18.7 |
| | | RMB ≥ 3501 (Income 4) | 210 | 6.9 |
| Travel mode share | Mode | Walk (Mode 1) | 970 | 32.1 |
| | | Moped (Mode 2) | 537 | 17.8 |
| | | Bus (Mode 3) | 549 | 18.2 |
| | | Taxi (Mode 4) | 445 | 14.7 |
| | | Private car (Mode 5) | 523 | 17.3 |

## 4. Methodology

The survey travel behavior responses included scale data and the outcome of a set of contributing factors. To model such count data, a number of verification and analysis models can be considered. The methodology involves three main procedures: (a) A factor analysis to identify latent attitudes, (b) SEM to reveal the structural relationship between latent attitudinal variables and transportation utility and (c) a comparison between different groups.

Factor analysis includes a factor structure evaluation, an exploratory factor analysis (EFA) and a confirmatory factor analysis (CFA). Cronbach's Alpha reliability coefficient was used to test the reliability of the latent attitudinal scale and a factor analysis was used to evaluate the factor structure of the scale. The EFA explains most of the covariance between variables. On the basis of determining the content of the result factors by EFA, CFA can establish the relationship between the observed identifying variables and the latent variables measured and enables an explicit test of the hypotheses about the data factor structure [38].

We used the information from the factor analysis to determine the model structure in the SEM. This study adopted the SEM approach, which involves two components: a measurement model and a structural model to explore choice behavior in small underdeveloped cities [39]. Figure 1 shows the SEM constructed for this study. The measurement model describes how well the observed indicators measure the latent variables and the structural model is used to relate all of the variables (both latent and observed).

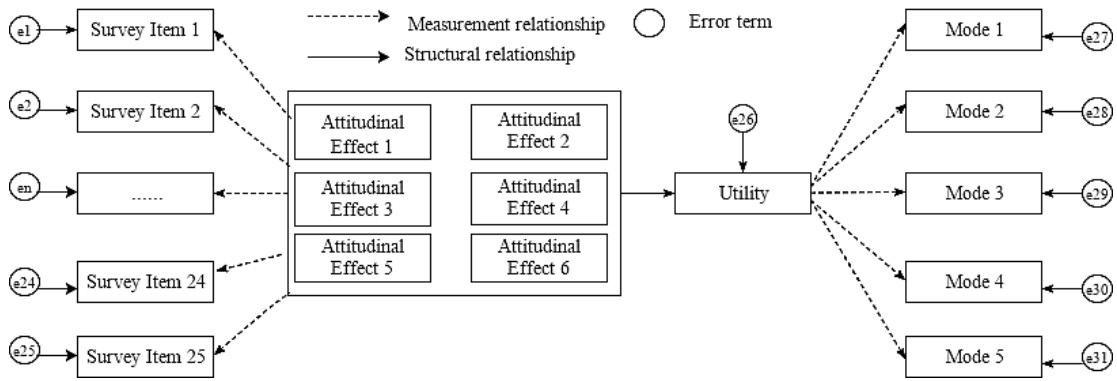

**Figure 1.** Example of the structural model examining mode choices behavior.

There are two main groups of observational variables: (a) attitude statement variables, which comprise survey Item 1 to Item 22 and (b) traffic mode use variables, which comprise Mode 1 to Mode 5 are the modes that each respondent actually used in the one-day travel record. There are three groups of latent variables: (a) Attitudinal Effect 1 to Attitudinal Effect 6, representing the most important preferences in travel behavior, (b) the "utility" variable, representing the transportation utility of the five types of traffic modes and (c) the error terms associated with each variable. The travel modes represent different transportation utilities for five types of transportation modes, including: pedestrian transportation utility, moped transportation utility, public transportation utility, taxi transportation utility and private transportation utility. Therefore, the travel modes represent the respondent's travel intentions, which are used to measure their transportation utility. In other words, the types of traffic modes represent the corresponding categories of transportation utility. The relationship between variables is represented by lines. A one-way arrow indicates an effectual relationship between two variables; the variable indicated by the arrow is affected by the other variable. Dashed lines correspond to the measurement equations and solid lines represent structural equations as in Figure 1.

The comparison between different groups includes a cluster analysis and a difference analysis. The cluster analysis was used to segment the public bus users based on the attitudinal factor scores from SEM. The difference analysis was used to analyze differences in travel preferences between groups with different sociodemographics. In order to refine the respondents' group travel attitude preferences, the k-means clustering approach was used to classify homogeneous groups in this study. The k-means clustering approach is a process of repeating the center point of a mobile class, moving the center point of the class—also called the center of gravity—to the average position of its containing members and then re-dividing its internal members. Conversely, to determine the differences between economic and social groups, statistical verification was necessary by comparing the obtained statistical verification values with the probability distributions of some random variables. The F value and the T value are these statistical verification values and the probability distributions to which they correspond are the F distribution and the T distribution. When $p < 0.05$, it is considered that the test results reach a significant level and there is a significant difference between the two independent samples.

## 5. Results and Discussion

### 5.1. Reliability Analysis

The Cronbach's Alpha coefficient must be greater than 0.7 for the variable to be considered to have good reliability [40]. If the Cronbach's Alpha coefficient increases after deleting an item, then those items with the Corrected Item-Total Correlation (CITC) less than 0.5 can be deleted. This method of eliminating variables can improve reliability and was also the basis for this study to exclude items [41].

The Cronbach's Alpha coefficient was improved after Questions A4, B5 and C4 were deleted and the final results of reliability analysis are presented as follows: (i) Cronbach's Alpha coefficients of the variables studied in terms of comfort preference, convenience preference, reliability preference, flexibility preference, safety awareness and environmental awareness are 0.793, 0.879, 0.833, 0.858, 0.872 and 0.886, respectively, (ii) the minimum CITC of items is 0.585. The results indicate that the measurement items have a high level of reliability.

### 5.2. Exploratory Factor Analysis

First, Kaiser-Meyer-Olkin (KMO) and Bartlett's spherical test were used as an initial strategy to analyze the distribution of data and the independence of variables. The KMO = 0.879 and the value of Bartlett's spherical test is 36,747.064 with the Sig. < 0.001, indicating that the questionnaire data meets the premise requirements of factor analysis.

Then, EFA, using principal component analysis, was applied multiple times for factor extraction, using a factor root greater than 1 to extract the common factor and the variance maximum orthogonal to perform the factor rotation. The results of the analysis are shown in Table 3.

When applying EFA to each group of indicators, factor loadings are the correlation coefficients between indicators and latent factors. The results in Table 3 show that the items underlying each indicator with a factor loading greater than 0.5 can be retained and it also means that all indicators contribute to the construction of latent preferences. The percentage of variance is the ability of index variables to interpret latent variables. This reveals the explanatory ability of latent variables with respect to indicators. The higher the percentage of variance, the more explanatory the indicator. The percentages of variance for the six factors are: 9.876%, 13.584%, 10.317%, 13.023%, 13.386% and 13.358%, respectively. The cumulative interpretation ability reached 73.544%, indicating that the scale has good structural validity.

**Table 3.** Results of EFA and descriptive statistics.

| Latent Variable | Items | Factor Loading | | | | | |
|---|---|---|---|---|---|---|---|
| | | 1 | 2 | 3 | 4 | 5 | 6 |
| Comfort preference | A1 | | | | | | 0.832 |
| | A2 | | | | | | 0.865 |
| | A3 | | | | | | 0.784 |
| Convenience preference | B1 | 0.753 | | | | | |
| | B2 | 0.869 | | | | | |
| | B3 | 0.839 | | | | | |
| | B4 | 0.826 | | | | | |
| Reliability preference | C1 | | | | | 0.79 | |
| | C2 | | | | | 0.834 | |
| | C3 | | | | | 0.834 | |
| Flexibility preference | D1 | | | | 0.737 | | |
| | D2 | | | | 0.744 | | |
| | D3 | | | | 0.860 | | |
| | D4 | | | | 0.873 | | |
| Safety preference | G1 | | 0.761 | | | | |
| | G2 | | 0.783 | | | | |
| | G3 | | 0.845 | | | | |
| | G4 | | 0.855 | | | | |
| Environmental preference | H1 | | | 0.837 | | | |
| | H2 | | | 0.824 | | | |
| | H3 | | | 0.815 | | | |
| | H4 | | | 0.72 | | | |
| Eigen value | | 6.980 | 2.383 | 1.966 | 1.840 | 1.615 | 1.396 |
| % of Variance | | 13.584 | 13.386 | 13.358 | 13.023 | 10.317 | 9.876 |
| Cumulative % | | 13.584 | 26.97 | 40.328 | 53.351 | 63.668 | 73.544 |

Note: T-statistics are shown in parentheses. Blank cells indicate no value for indicator in column.

*5.3. Confirmatory Factor Analysis*

The CFA was used to test the convergence validity of the internal items of each variable. The main purpose of this test was to verify the adaptation of the actual measurement data to the theoretical framework. Based on the preliminary EFA, six latent attitudinal factors were tested for convergence validity through CFA.

The results of the validity test were evaluated according to some statistics, with their recommended values in parentheses [42], namely, (i) the minimum Chi-square (CMIN) with degrees of freedom (DF), (CMIN/DF, less than 5.0 and for a large sample size less than 10.0), (ii) the goodness-of-fit index (GFI, above 0.90), (iii) the adjusted goodness of fit index (AGFI, above 0.90), (iv) the normed fit index (NFI, above 0.90), (v) the incremental fit index (IFI, above 0.90), (VI) the Tucker-Lewis index (TLI, above 0.90) and (vii) the comparative fit index (CFI, above 0.90). The model fitting parameter indexes are: CMIN/DF = 8.892, GFI = 0.950, AGFI = 0.934, NFI = 0.953, IFI = 0.958, TLI = 0.950, CFI = 0.958, providing evidence of a proper fit.

For identification, the first indicator of each factor was selected as the base, as shown in Table 4. The convergence validity of the internal items of each variable was evaluated as follows [31,32]. First, the factor loading of each item was each greater than the threshold value 0.50. Second, the composition reliability (CR), indicating the internal consistency of the facet problem was larger than the recommended value 0.70. Third, the Average Variance Extracted (AVE)—the ability to calculate the variation of the variable for each measurement subject—value ranged from 0.575 to 0.668, all greater than the recommended value 0.50, showing a reasonable validity of the measurement model.

**Table 4.** CFA for the measurement model.

| Latent Variable | Items | Non-standardized Factor Loading | S.E. | C.R. | P | Standardized Factor Loading | CR | AVE |
|---|---|---|---|---|---|---|---|---|
| Comfort preference | A1 | 1 | - | - | - | 0.726 | | |
| | A2 | 1.067 | 0.031 | 34.804 | *** | 0.864 | 0.801 | 0.575 |
| | A3 | 0.788 | 0.024 | 32.892 | *** | 0.672 | | |
| Convenience preference | B1 | 1 | - | - | - | 0.727 | | |
| | B2 | 1.73 | 0.038 | 45.697 | *** | 0.874 | | |
| | B3 | 1.625 | 0.038 | 43.326 | *** | 0.822 | 0.885 | 0.66 |
| | B4 | 1.688 | 0.039 | 43.13 | *** | 0.818 | | |
| Reliability preference | C1 | 1 | - | - | - | 0.695 | | |
| | C2 | 1.342 | 0.034 | 39.07 | *** | 0.838 | 0.838 | 0.635 |
| | C3 | 1.198 | 0.031 | 39.204 | *** | 0.848 | | |
| Flexibility preference | D1 | 1 | - | - | - | 0.688 | | |
| | D2 | 1.087 | 0.032 | 34.456 | *** | 0.692 | | |
| | D3 | 1.64 | 0.039 | 41.713 | *** | 0.871 | 0.863 | 0.615 |
| | D4 | 1.547 | 0.037 | 41.572 | *** | 0.866 | | |
| Safety consciousness | E1 | 1 | - | - | - | 0.680 | | |
| | E2 | 1.364 | 0.037 | 36.936 | *** | 0.750 | | |
| | E3 | 1.556 | 0.038 | 41.463 | *** | 0.863 | 0.875 | 0.639 |
| | E4 | 1.645 | 0.039 | 42.136 | *** | 0.886 | | |
| Environment awareness | H1 | 1 | - | - | - | 0.890 | | |
| | H2 | 1.027 | 0.016 | 65.728 | *** | 0.889 | | |
| | H3 | 0.847 | 0.017 | 49.629 | *** | 0.747 | 0.889 | 0.668 |
| | H4 | 0.789 | 0.017 | 47.785 | *** | 0.729 | | |

*** $p < 0.001$.

The discriminant validity analysis was applied to verify whether there were statistical differences between the two different facet correlations. The values of the square root of the AVE for each factor on the diagonal were larger than the squared correlations outside the diagonal in Table 5, so we concluded that the discriminant validity is adequate for the measurement model [43,44].

**Table 5.** Squared correlation matrix of factors.

| Latent Variable | A | B | C | D | G | H |
|---|---|---|---|---|---|---|
| A | **0.758** | - | - | - | - | - |
| B | 0.271 ** | **0.812** | - | - | - | - |
| C | 0.059 ** | 0.304 ** | **0.796** | - | - | - |
| D | 0.122 ** | 0.270 ** | 0.322 ** | **0.784** | - | - |
| G | 0.182 ** | 0.295 ** | 0.335 ** | 0.355 ** | **0.799** | - |
| H | 0.214 ** | 0.408 ** | 0.425 ** | 0.384 ** | 0.418 ** | **0.817** |

Note: The numbers on the diagonal are AVE, while the off-diagonal values are squared correlations between factors.
** $p < 0.01$.

### 5.4. SEM on the Whole Sample

Through the prior information of the correlation structures in the above analysis, the model structure shown in Figure 2 was specified in SEM. The measurement model and the structure model were calibrated simultaneously in SEM. The SEM results, including the factor loadings of statement variables on latent attitudinal factors and the coefficients of correlations between those latent factors, are shown in Figure 2. The fit indices are CMIN/DF = 8.492, GFI = 0.950, AGFI = 0.934, NFI = 0.952, IFI = 0.958, TLI = 0.949 and CFI = 0.958, such that the model provides an appropriate fit to the sample.

The total effects of the latent variable on the target transportation utility are summarized in Table 6 to provide an intuitive understanding of the psychological factors that each plays. The primary conclusions based on Figure 2 and Table 6 are drawn as follows: (i) at the 95% confidence level, respondents' latent attitudinal variables are positively associated with the transportation utility, indicating that respondents are more likely to be satisfied with this mode, (ii) transportation utility was explained by six latent attitudinal variables including: comfort preference, convenience preference, reliability preference, flexibility preference, safety consciousness and environmental awareness,

(iii) among all the factors that influence transportation utility, flexibility preference (0.07) and environmental awareness (0.07) had the least effect. In other words, most residents tend to participate in subsistence activities, which demands less flexibility when compared to recreational activities. Likewise, compared with large cities, the ecological environment of small cities is generally better. As such, it is understandable that respondents are less likely to give importance to environmental issues in travel. Finally, (iv) safety consciousness (0.12) had the greatest effect on the transportation utility, followed by convenience preference (0.11), reliability preference (0.10) and comfort preference (0.09). These four latent attitudinal variables are more likely to change travel behavior in small, underdeveloped cities.

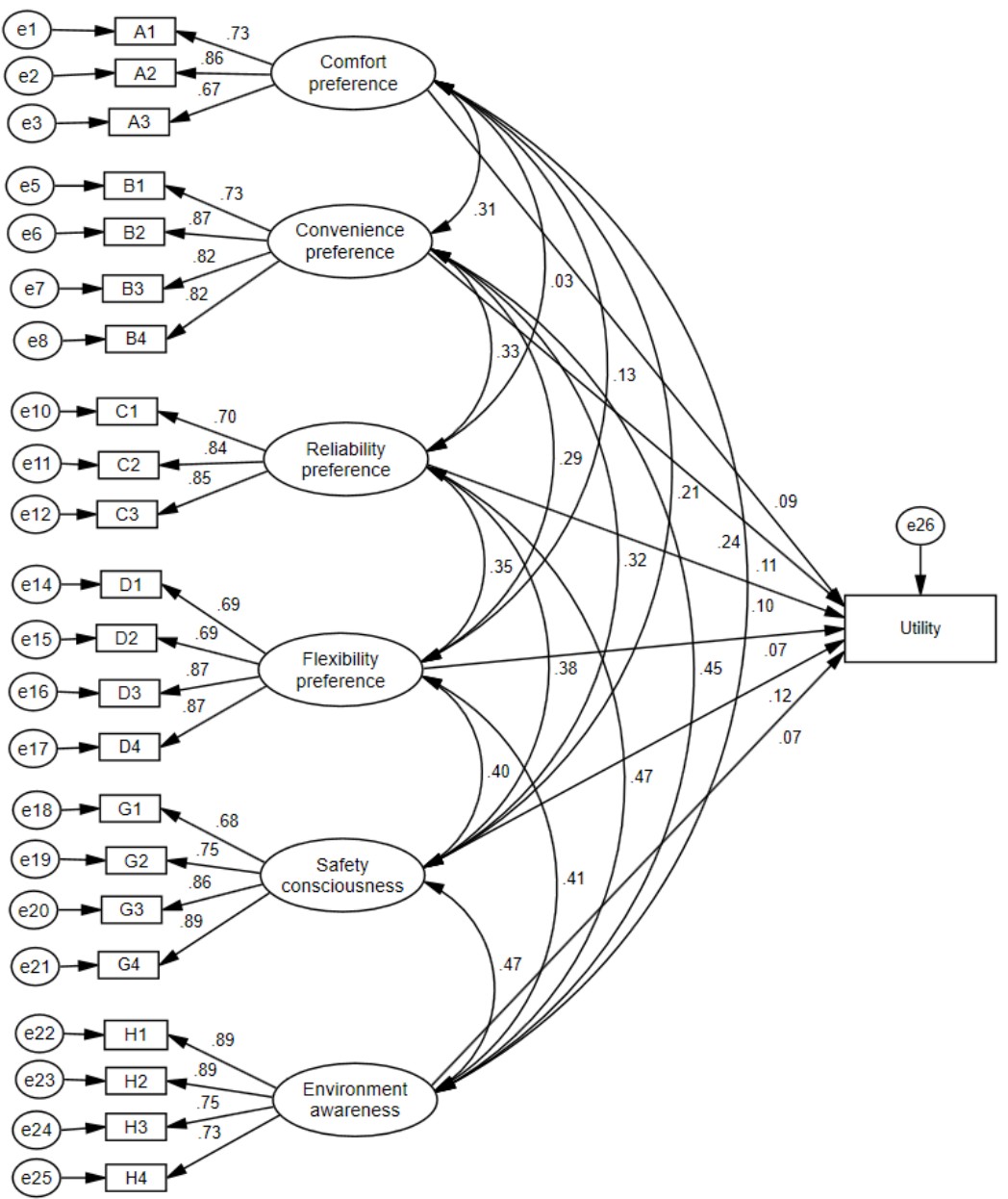

**Figure 2.** Calibration results of the SEM model.

**Table 6.** The effects of latent variables on transportation utility.

| Latent Variable | Transportation Utility | Effect | S.E. | C.R. | P |
|---|---|---|---|---|---|
| Comfort preference | Travel behavior | 0.09 | 0.037 | 4.873 | *** |
| Convenience preference | | 0.11 | 0.058 | 4.202 | *** |
| Reliability preference | | 0.10 | 0.046 | 4.139 | *** |
| Flexibility preference | | 0.07 | 0.047 | 3.133 | 0.002 |
| Safety consciousness | | 0.12 | 0.057 | 5.332 | *** |
| Environmental awareness | | 0.07 | 0.038 | 3.274 | 0.001 |

*** $p < 0.001$

### 5.5. Comparison between Different Groups

According to this research, four attitudinal factors—safety awareness, desire for comfort, need for reliability and demand for convenience—were chosen to classify similar attitudinal groups. Subsequently, k-means clustering analysis was carried out for different clustering solution groups and five clustering solutions were selected for further analysis. In the process of the cluster analysis of respondents, each cluster was divided into public transportation users and non-public transportation users (e.g., moped users, private car users). Some non-public transportation users are potential customers of public transportation and those potential customers are also targeted by public transportation. The results of clustering which produce the most possible explanation were used to describe the differences between clusters. The results that show distinct characteristics among clusters are presented in Table 7.

**Table 7.** Segmentation Result for Public Bus Users.

| Attitudinal Factor | C1 (162/731) | C2 (71/387) | C3 (141/839) | C4 (85/588) | C5 (90/479) |
|---|---|---|---|---|---|
| Safety awareness | 3.16 £ | 2.38 | 3.42 £ | 2.11 | 2.53 |
| Desire for comfort | 2.30 | 3.34 £ | 3.56 ※ | 1.78 | 3.50 ※ |
| Need for reliability | 3.27 £ | 1.91 | 3.61 ※ | 2.23 | 2.03 |
| Demand for convenience | 3.21 £ | 1.85 | 3.38 £ | 2.10 | 3.77 ※ |

Note: ※ and £ indicate intervals in which the cluster center belongs to high and moderate levels, respectively. Values indicate a cluster center in one dimension of an attitudinal factor, high level (>3.5), moderate level (2.6–3.5) and low level (<2.6). Values in parentheses indicate the number of bus trips and the total number of clusters, respectively.

The attributes of attitudinal factors within each cluster can be summarized as follows: (i) Cluster 1 (C1) is a group with moderate safety awareness and a moderate need for reliability and convenience; (ii) Cluster 2 (C2) is a group with a moderate desire for comfort; (iii) Cluster 3 (C3) is a group with moderate safety awareness, a moderate demand for convenience, yet a high desire for comfort and convenience; and (iv) Cluster 5 (C5) is a group with a high desire for comfort and convenience. C1 and C5 have a relatively higher percentage of public bus users (22% and 18.7%, respectively). Conversely, the percentage of public bus users in C4 is the lowest. Among the five clusters, the proportion of public bus users of C2 (18%), C3 (16.8%) and C4 (14.5%) was lower than that of all the respondents (18.2%). C3 is a group with a high desire for comfort and need for reliability, yet moderate demand for convenience and safety awareness and, as such, public transportation has difficulties in meeting these inner preferences. In addition, C2, with a moderate desire for comfort and the residents of C2 and C4 have no obvious requirements in the four inner preferences. There is no obvious tendency for the two groups to choose public transportation. As a result, residents in C1 and C5 can be easily persuaded to become public bus users and residents in C2 and C4 also have the potential to switch to public transit.

The k-means clustering analysis in this study helps identify respondents with homogeneous attitudes toward transportation utility. Table 8 shows the differences in travel preferences among groups with different sociodemographics and requires focus on the process of implementing travel demand management. The explanatory variables—gender, age and income—were extracted from

the sociodemographics because they had statistically significant differences in at least four attitudes ($p$-value < 0.05).

**Table 8.** Travel preferences among groups with different sociodemographics.

| Variable | Average Score | | | | | |
|---|---|---|---|---|---|---|
| | **A** | **B** | **C** | **D** | **G** | **H** |
| Gender 1 | 2.827 | 2.993 | 2.717 | 2.901 | 2.779 | 2.810 |
| Gender 2 | 2.913 | 2.918 | 2.874 | 3.044 | 2.876 | 2.989 |
| T/P | 2.668/0.008 | 2.298/0.022 | −4.602/0.000 | −4.185/0.000 | −3.092/0.002 | −5.419/0.000 |
| Age 1 | 2.583 | 2.929 | 2.920 | 2.857 | 2.879 | 2.951 |
| Age 2 | 2.975 | 3.063 | 2.896 | 3.135 | 2.865 | 2.939 |
| Age 3 | 2.815 | 2.987 | 2.794 | 3.012 | 2.830 | 2.925 |
| Age 4 | 3.026 | 2.857 | 2.629 | 2.908 | 2.778 | 2.789 |
| Age 5 | 3.111 | 2.913 | 2.682 | 2.907 | 2.747 | 2.831 |
| F/P | 30.902/0.000 | 4.118/0.002 | 9.570/0.000 | 7.546/0.000 | 2.038/0.086 | 3.349/0.010 |
| Income 1 | 2.742 | 2.923 | 2.849 | 3.013 | 2.864 | 2.944 |
| Income 2 | 2.847 | 3.012 | 2.796 | 2.915 | 2.795 | 2.893 |
| Income 3 | 3.061 | 2.940 | 2.657 | 3.014 | 2.791 | 2.782 |
| Income 4 | 3.238 | 2.958 | 2.806 | 2.842 | 2.833 | 2.936 |
| F/P | 31.243/0.000 | 1.888/0.129 | 5.490/0.001 | 3.667/0.012 | 1.569/0.195 | 4.294/0.005 |

Male respondents had greater convenience preferences than women in travel, while other requirements were lower for men than female respondents. In small, Chinese cities, adult women are more involved in family affairs, such as fetching children and caring for the elderly, so they have higher requirements for flexibility. Women also pay more attention to comfort while traveling. For example, if carrying a simple parcel or shopping bags, women may require a seat to relieve fatigue. In terms of comfort, Age 5 (≥65 years old) scored the highest, followed by Age 4 (55–64 years old), which is consistent with the natural laws of human aging. Age 1 group (<18 years old) was mainly students. Students not only need to know the departure time of the transportation but also need to control the time required for the entire trip to arrive at school on time. As a result, this group has the highest reliability score and the daily travel of most people in this group was relatively fixed, especially from Monday to Friday. In addition to participating in subsistence activities, Age 2 (19–34 years old) and Age 3 (35–54 years old) also tended to travel for maintenance and discretionary activities. The diversity of the two groups' participation needs to match the flexibility in their travel processed, so the flexibility scores of the two groups were 3.135 and 3.012, much higher than Age 1, which had the lowest flexibility score, 2.857. The group under Income 1 (RMB ≤ 1499) paid more attention to the reliability of the travel mode, followed by the group with the highest income (Income 4; RMB ≥ 3501). Although these two groups of people have different socioeconomic attributes, they all require reliability from the perspective of time value. The Income 3 group (RMB 2500–3500), which have greater social activities, are generally 25-54 years old in small cities. This group needs to deal not only with personal matters (e.g., fitness, continuing education, etc.) but also with family affairs (e.g., fetching children or transporting the elderly to hospital). The Income 3 group has a strong purchasing power and therefore needs a flexible travel experience for shopping, consistent with residents of large cities. In small cities, Income 1 is primarily a student group and Income 4 are generally highly educated. These two income-level groups pay more attention to the environment and are more willing to change their mode of travel.

## 6. Summary and Conclusions

This study aimed to understand travel behavior in small underdeveloped cities in China and reveals the effects of latent attitudinal variables on travel model choices. In addition, we examined differences in sociodemographics on travel modes. The paper's authors have worked at the Tibet University and study at Jilin University, both of which are located in underdeveloped areas in China.

The authors have facilitated long-term, intermittent observation and research on local resident lifestyles and actively promote fair access to travel and development of transportation. This study used residents trip survey data from two small cities in different regions of southwest and northeast China. To that end, a structural equation model was applied to analyze the impact of latent attitudinal variables on travel effectiveness in different ways. We also explored the differences between varying sociodemographics. The analysis yielded some interesting conclusions.

The first conclusion is that structural equation models with latent variables (comfort preference, convenience preference, reliability preference, flexibility preference, safety consciousness and environmental awareness) are useful elements in understanding residents' personal preferences and travel behavior in underdeveloped areas. Therefore, this study supports the adjustment of the city's travel structures by satisfying residents' personal attitude preferences.

According to the selected attitude factors, the k-means clustering method was used to divide respondents into several clusters with distinct intrinsic preferences and each cluster was distinguished by public transport users and non-public transport users. The results show that the attitude-based clustering method can effectively capture the heterogeneity of public bus users. In this study, respondents were divided into five clusters through four attitude factors: safety awareness, desire for comfort, need for reliability and demand for convenience. Different clusters show obvious attitudinal preferences in public bus users.

C1 is a group with a moderate demand for convenience, reliability and safety. C5 is a group with a high desire for comfort and convenience. The proportion of public transport passengers in C1 and C5 is higher than that of all respondents. However, the ratio of C2, C3 and C4 is lower than average. C3 is a group with a high desire for comfort and a need for reliability. The residents of C2 and C4 have no obvious requirements in the three preferences, including convenience, reliability and safety. Public transportation has clear difficulties in attracting non-public transport users in C3 because their preferences have higher importance. Non-public transport users in C1 and C5 can be more easily persuaded to use bus transit and non-public transport users in C2 and C4 have the potential to switch to public transit. Therefore, future active transport policy should focus on meeting the travel preferences of C1 and C5 in order to increase bus utilization.

We also examined differences in sociodemographics, including family attributes (e.g., family size, number of cars and number of motorcycles/bicycles), individual attributes (e.g., sex, age, disposable income and occupation type) and trip attributes (e.g., trip chain type and travel purposes) that could affect attitude preference. We conclude that people of different ages, genders and with different levels of disposable income have greater differences in attitudinal preferences of comfort, convenience, reliability and flexibility. Key groups for consideration are women, low-income groups and the elderly. Different groups have a unified understanding of safety in travel and many respondents have high requirements for safety.

In addition, in small, underdeveloped cities in China, residents are not paying enough attention to environmental awareness. At present, compared to large cities in China, the air quality of smaller cities is one of the important reasons for some respondents choosing to place less importance on environmental protection. Furthermore, as residents grow older, their demands for comfort will gradually increase, which reflects the natural law of human aging. Respondents in the 1–18 age group were mainly students, who had the highest requirement for travel reliability because of the importance of getting to school on time. In addition to participating in travel for subsistence activities, people aged 19–34 and 35–54 also need to participate in maintenance and discretionary activities. Therefore, flexible travel modes are more aligned with the travel attitude of such groups.

Many factors influence people's travel behavior and many of these factors are not easy to change in small, underdeveloped, Chinese cities. In small, underdeveloped cities, we recommend that government implements public transportation policy guided by the four attitudes—safety, reliability, convenience and comfort—to guide residents in achieving efficient and low-carbon travel. Policy-makers should pay more attention to actions improving safety and reliability, including the

overall monitoring of bus dynamic operation and providing arrival times on electronic stop boards or mobile phone software. In addition to increasing bus routes, the government also needs to increase its appeal to the elderly and low-income groups (e.g., through free tickets, monthly passes and student discounts). Furthermore, priority should be given to female passengers carrying parcels, such as additional priority seats on buses. The results of this study have important implications for policy development to better meet the psychological travel preferences of residents, which can help reduce the dependence on private cars and effectively alleviate the increasingly prominent traffic congestion problems in small cities.

**Author Contributions:** G.C. and S.Z. proposed the idea and designed the survey; G.C. analyzed the data and wrote the article; J.L. aided in the improvement of the figures and manuscript. All authors read and approved this version.

**Funding:** This research was supported by the National Social Science Foundation of China (Grant No. 18BSH077).

**Acknowledgments:** We extend our thanks to the Bus Company of the Lhasa Public Traffic Group and the Transportation Bureau of Yushu for valuable data.

**Conflicts of Interest:** The authors declare no conflict of interest.

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
