# Peer review of "The Effects of Latent Attitudinal Variables and Sociodemographic Differences on Travel Behavior in Two Small, Underdeveloped Cities in China"

_sustainability, doi:10.3390/su11051306_

Round 1

Reviewer 1 Report

p.p1 {margin: 0.0px 0.0px 0.0px 0.0px; font: 12.0px 'Helvetica Neue'; -webkit-text-stroke: #000000} p.p2 {margin: 0.0px 0.0px 0.0px 0.0px; font: 12.0px 'Helvetica Neue'; -webkit-text-stroke: #000000; min-height: 14.0px} li.li1 {margin: 0.0px 0.0px 0.0px 0.0px; font: 12.0px 'Helvetica Neue'; -webkit-text-stroke: #000000} span.s1 {font-kerning: none} ul.ul1 {list-style-type: none}

This is an interesting paper that aims to investigate the influence of latent attitudes and intentions on travel behavior, for small cities in China. This is important, given that the large volume of study on travel behavior tends to focus on large, dense, and populous areas. Although this study contributes some new knowledge to the field of travel behavior analysis, I have the following comments on the paper content and suggestions/questions around improving the discussions of the results:

Survey:

The paper provides descriptive statistics on the survey respondents, but provides little discussion of how well the samples represent the populations of the cities. This is a missed opportunity, given the aim of generalizing the results for small cities in China.

Utility Modeling:

The paper does not detail the mode utility specification process nor estimation results. It simply refers to this as the “transportation utility”. I am assuming that some discrete choice analysis process was conducted, given that there was mention of the effects of latent variables on utility, and given that the Literature Review section starts out with a discussion on Random Utility Modeling? If so, what were the utility structures and justifications for these structures? What were the estimation results? 

Language Use:

The language used throughout the paper is generally very technical and full of jargon, with little interpretation provided, nor discussion of the intuition given. For example, the results section details various statistical tests performed on the latent variables, but there are no justifications or further discussions provided. Perhaps it would be helpful if these methods for validating/verifying the latent variables were better discussed in the Methodology section.

The authors in some cases make references to the variable code names (e.g. Age 4) rather than the meanings/ representations of the variables (respectively, 55-60 year olds), which forces the audience to keep going back to reference tables in order to follow the discussion.

Statements like the following: "Shopping is one of the most effective ways for urban residents to relieve stress.” gives me the impression that the authors  are grasping at ways to explain the behavior, rather that using the survey data or statistics about the city to explain behavior.

Paper Contributions:

While the application of latent variable models on this new dataset is offered at a contribution, it is not clear what the combination of latent variable and SEM reveal, that could not be otherwise revealed from basic multinomial logic? It is important to discuss the advantages in explanatory power offered by using more sophisticated modeling tools, in this case.

Author Response

Thank you for your kind consideration and valuable comments. Your comments have helped us to make out manuscript more logical, systematic, and reasonable, particularly, the Utility Modeling and Language Use sections. This will also be useful for writing future papers.

Reviewer 2 Report

The paper gives a good overview about the upcoming challenges in underdeveloped cities in China and can be regarded as a resource paper for others with its conceptual intention. I would suggest saying more about the motivation at the beginning of the introduction. What is the main problem?

In the introduction, however, the authors pointed out that by answering the research question, the link between "travel behaviour" and " Latent Attitudinal Variables" in two small cities. But why aren't the analyses differentiated according to the two or input into the cluster method?

The paper has good elements and includes its own survey. As far as research design and data collection and evaluation are concerned, the study has for the most part been well and comprehensibly structured.

The paper shows a lot of relevant literature. It would be fine.

The survey methodology is empirical state of the art.  Methodically it works reputable. The methodical part is also much more fluid than the theoretically part. the research question is clearly formulated in the introduction, has been comprehensively investigated in the methodical part, and has been answered in the conclusion.

The results presented clearly – especially some data analysis and the visualization have to be improved. Neither the key parameters of the mobility like modal split, number of trips, travel time per day etc in Yusho or Lhasa are missing – also the difference of the travel behaviour between a small and a big city in China.  A little table would be fine.

Page 234 line 228: Can you explain stable career more detail?

Table 2. Why are the results not separated by the cities? Maybe it would be very interesting to know it. Are Trip distance and trip duration part of the survey? It´s not clear, what is the travel mode share-  a typical day, a reported day, etc.

Are you using a weighting factor to adjust out socio demographic skewness?  

5.4 SEM: What is the unit of the utility function in the SEM modelling? It should be describes more detail.

Line 356: Figure 2:  I can´t follow the expression calibration

5.5 Clustering: Table 7 shows the result only for PT – what is the result of the other traffic modes?

Table 8: There is always no determination between both cities / Variables like travel mode choice are not listed.

Nevertheless, the conclusions remain a little bit unspecific (Who it addressed?). Perhaps the conclusions can be sharpened in terms of "mode choice and SEM". Why need urban and traffic planner such results. Is it really true, that latent variables improve traffic models for planning processes?

Author Response

We thank you for your time and efforts in reviewing this paper. Your comments have helped us to make out manuscript more logical, systematic, and reasonable.We have carefully addressed each comment and the corresponding amendments are updated in the revised manuscript. 
